# Linkage Disequilibrium Decay in Selected Cattle Breeds

**DOI:** 10.3390/ani14223317

**Published:** 2024-11-18

**Authors:** Farhad Bordbar, Just Jensen, Armughan Ahmed Wadood, Zipei Yao

**Affiliations:** 1State Key Laboratory of Livestock and Poultry Breeding, Lingnan Guangdong Laboratory of Agriculture, Guangdong Provincial Key Lab of Agro-Animal Genomics and Molecular Breeding, Key Lab of Chicken Genetics, Breeding and Reproduction, Ministry of Agriculture and Rural Affair, South China Agricultural University, Guangzhou 510642, China; armughanwadood@gmail.com (A.A.W.); datuohai@stu.scau.edu.cn (Z.Y.); 2Center for Quantitative Genetics and Genomics, Aarhus University, 8000 Aarhus, Denmark; just.jensen@qgg.au.dk

**Keywords:** linkage disequilibrium, genome SNPs data, cattle, Nellore, Sahiwal

## Abstract

The development of new instruments, such as dense genetic markers, has led to a resurgence of interest in linkage disequilibrium analyses. Alleles at two or more loci that are not randomly associated are referred to as LD. In this research, LD values accurately represented both a genetic similarity and varying historical backgrounds in different cattle breeds. The Sistani breed has an LD decay like indicine cattle (Nellore and Sahiwal) in comparison with other breeds. The single-nucleotide polymorphism (SNP) density used in this study will enable the implementation of genomic selection programs and whole genome studies in native cattle in different regions of the Middle East.

## 1. Introduction

Iran has a long history of domesticating and rearing cattle. Iranian native cattle have been raised throughout several geographic areas in the nation and, therefore, have adapted to a variety of environmental circumstances. Among them, native to the region, Sistani beef cattle are a breed of Bos indicus that has adapted to the hard climate including resistance to heat and drought [1]. This breed is mostly found in Sistan and Baluchestan province close to the Iran–Pakistan border.

Linkage disequilibrium (LD) is defined as the non-random relationship between loci. There has been a renewed interest in LD due to the emergence of new tools like dense genetic markers. LD means simply a nonrandom association between alleles at two or more loci. Genomic selection (GS) and genome-wide association studies (GWAS) are theoretically based on LD, which is also relevant for gene mapping, effective population size estimations, population structure analysis, genomic and population parameter estimation, and other related fields [2,3]. The extent of LD across genomic regions, using LD information between markers and quantitative trait loci (QTLs), is an important parameter to determine the statistical power of GWAS with single-nucleotide polymorphisms (SNPs) [4]. In natural populations, LD is affected by genetic drift, migration, selection, mutation, and recombination [5,6]. An increasing number of studies have aimed at quantifying LD characteristics in domestic animals [7], especially in cattle [8,9].

Most studies used a low marker density or were performed in limited regions of the studied genomes. Farnir et al. [10] performed the first whole-genome LD study to characterize the extent and pattern of LD based on the information of 284 microsatellite markers in Dutch Holstein cattle. Several subsequent studies have confirmed extensive LD in cattle [11,12]. The average LD with increasing distance between markers appears to vary according to population structure and parameters such as effective population size [11,12].

Along with access to many SNP markers throughout the genome, several studies have assessed the pattern of LD in cattle breeds [13,14,15,16,17,18,19]. LD mapping tools are very important for exploring economically important traits in cattle using GWAS and/or genomic prediction models. The comparison of LD maps may be used to study the genetic basis of economically significant traits [20], the identification of genomic areas under selection pressure [5], and breed variety in cattle [11]. Furthermore, the degrees of LD in the populations under study are impressive in terms of their ability to improve the efficiency of several other regular investigations used in animal breeding, such as parentage testing, GWAS, and QTL mapping [11,21].

The breeding and domestication of cattle in Iran has historical origins. Iranian indigenous cattle are scattered in different geographic regions and great variety in appearance is clear [22]. In a study recently started using SNP markers on Iranian indigenous cattle breeds (Mazandarani, Talesh, Kurdi, Nejadi, and Sarabi), it was shown that there are genetic diversity and different historical origins in Iran’s domestic cattle breeds [22]. Currently, the number of breeds in these areas are mixed and there is no selection for increased production. These cows appear mostly black and have poor milk production. Both sexes have horns and their ears hanging half in comparison to other breeds in the region. This breed has small organs with elegant bones and weak muscles and has a small bulk compared to other breeds. In recent years, the number of indigenous cattle has been declining rapidly across the country. The aim of this study was to determine the extent of linkage disequilibrium in the genome of indigenous cattle in the Sistan and Baluchestan province and compare with two subspecies of Bos indicus cattle based on dense SNP data. The density required for performing procedures such as genomic selection and/or whole-genome association studies in this population will also be investigated.

## 2. Material and Methods

### 2.1. Sample Collection and Genotyping

For this study, twelve samples from each of the five breeds were collected, i.e., a total of sixty individuals were included. The study’s animals comprised the three primary breeds of cattle: Bos indicus breeds: (Sahiwal (dairy) and Nellore (beef)), Bos taurus: (Hereford (beef) and Holstein (dairy)) and indigenous cattle (Sistani) from the Sistan and Baluchestan province. The data collection of Sahiwal, Nellore, Hereford, and Holstein cattle was located in Punjab province, Pakistan. The BovineHD SNP chip, developed by Illumina, was utilized to genotype samples. This chip can identify 777,962 SNPs.

### 2.2. Quality Control, Scores, and Minor Allele Frequency

The quality control (QC) of the SNP data was conducted with PLINK 1.07 software [23]. Every locus on chromosomes X, Y, and mitochondrial DNA was disregarded. Hardy–Weinberg equilibrium in all loci was examined and all loci with a *p* < 10^−7^ were excluded. Minor allele frequency (MAF) less than 0.05 was excluded across the five breeds involved. Loci with call rates less than 90% were excluded. For quality score, ten samples (genetically sequenced cattle) were randomly chosen. Quality scores (Q), which show the quality and reliability of SNP data [4], were calculated based on the concordance rate (*Po*) determined by calculating the chance agreement (*Pc*) and the percentage of imputed SNPs that match an actual genotype [24]:$$Po=\frac{\sum inij}{n..}Pc=\frac{\sum \;ini.n.i}{n..^{2}}$$
where n represents the total number of individuals, and nij shows the number of individuals with accurate genotype *i* and true genotype *j*.

### 2.3. Estimation of Linkage Disequilibrium

The linkage disequilibrium between SNP pairs was measured using *D*′ and r^2^ on the same chromosomes [22]. The following equation was used:$$D=freq\;\left(A1-B1\right)\times freq\left(A2-B2\right)\_freq\left(A1-B2\right)\times freq\left(A2-B1\right)$$
where *freq A* and *B* are the frequencies of alleles *A* and *B*. $freq\left(A1-B1\right)\;\mathrm{and}\;\left(A2-B2\right)\;$ are the frequencies of the A1_B1 and A2_B2 haplotypes in the population. The *D* statistic is dependent on the frequencies of the individual alleles and is therefore not useful for LD among multiple pairs of loci. *D′* and *D* max were calculated as follows:$$\begin{array}{c} D^{\prime }=\frac{\left|D\right|}{D\;max}\\ \mathrm{If}\;0<D,\mathrm{then:}\\ D\;max=min\left[f\;\left(A1\right)\times f\;\left(B2\right)-1\times f\;\left(A2\right)\times f\;\left(B1\right)\right]\\ \mathrm{If}\;0>D,then,\\ D\;max=min\left[f\;\left(A1\right)\times f\;\left(B1\right)-1\times f\;\left(A2\right)\times f\;\left(B2\right)\right] \end{array}$$

The *r*^2^ statistic was defined as the correlation between SNP pairs; compared with *D′*, it shows greater stability and is less sensitive to fluctuations due to sample size and differences in allele frequencies [25,26], and it was calculated according to the following equation [27]:$$r2=\frac{{\left(freqAB\times freqab-freqAb\times freqaB\;\right)}^{2}}{\left(freq\;A\times freq\;a*freq\;B\times freq\;b\right)}$$

Applying the SnppldHD program (Sargolzaei, University of Guelph, Guelph, ON, Canada), the measures of LD (*r*^2^) were determined for each marker pair on each chromosome (syntenic SNPs). The calculation of *r*^2^ was assessed at distances of 1–50 Kb, 50–100 Kb, 100–500 Kb, and 0.5–1 Mb, and average *r*^2^ values for all autosomes were calculated within distance classes. Using the following equation, a sample size adjustment was applied to the calculated *r*^2^ values [28]:$$r^{2}\;corrected=\frac{r^{2\;}computed-\frac{1}{n}}{1-\frac{1}{n}}$$
where *n* is the number of alleles in the sample (the sample number is multiplied by two). All cattle breeds’ average *r*^2^ values and standard deviations, summed over all autosomes, were calculated in 1 Kb to 1 Mb. Statistical analysis was performed in GraphPad Prism (version 6.0, GraphPad Software, San Diego, CA, USA) and Excel 2016 with a significance criterion of *p* < 0.05. We also used SnppldHD for LD calculation. The application of this software enabled us to calculate LD for biallelic markers and, therefore, it is considered one of the most reliable software to calculate LD.

## 3. Results

For a total of 777,563 SNPs, 60 cows underwent genotyping. SNPs located on the mitochondrial, X, and Y chromosomes (48,088, 3813 and 991 SNPs, respectively) were not included in the analysis. In total, 296,125 SNPs that deviated from Hardy–Weinberg equilibrium at a *p* < 10^−7^ were deleted from the dataset, and 253,123 SNPs with call rates of less than 90% and a MAF of less than 0.05 were also eliminated. Principally, the 175,423 SNPs that were left following QC showed an overall MAF mean of 0.183. Table 1 represents the SNP number remaining following QC, the overall mean of MAF, the average SNP interval, and total SNP pairwise comparisons in each breed. The general QC was carried out across breeds and therefore some SNPs were found to be monomorphic in individual breeds. A total of 8294 monomorphic SNPs were discovered among the filtered SNPs: they included 1202 in Sistani, 1321 in Nellore, 1392 in Sahiwal, 2256 in Herford, and 2123 in Holstein (where monomorphic SNPs partially overlapped with each other). The average of quality score and concordance rate representing the reliability and quality of SNP data [24] recorded 0.93% (Figure 1). The quality scores of 10 randomly chosen samples are listed in Appendix A.

A noteworthy finding suggests a possible close relationship between the Sistani, Nellore, and Sahiwal populations: about 80.21% of monomorphic markers in the Sistani population were also monomorphic in the Nellore and Sahiwal populations. Figure 2 illustrates the histogram of MAF in the range from 0.0 to 0.5 in different breeds. The highest proportion of SNPs belong to Nellore and Sahiwal, with a MAF between 0 and 0.1, and the lowest proportion of SNPs belong to Nellore and Sahiwal, with a MAF between 0.4 and 0.5.

### 3.1. Sample Quality and Minor Allele Frequency Distribution

In this study, the amount of gametic phase disequilibrium obtained by r^2^ statistics for all breeds was compared. The lowest average values of minor alleles were in the Nellore breed, with a mean of (0.130), and the highest average values of minor alleles were in Holstein, with a mean of (0.252) (the impact of SNP ascertainment bias must be acknowledged). The Nellore breed had the largest percentage of SNPs with a MAF of less than 0.2 (74%), whilst the Holstein breed had the lowest percentage of this fraction (46%). The lowest MAF was estimated in the Nellore breed, and this breed also had the highest proportion of SNPs located in categories with a MAF < 0.2, alongside Sahiwal and Sistani, which showed the same trend.

### 3.2. The Gametic Phase Disequilibrium Rate Between Different Chromosomes

The corrected average r^2^ values of all SNPs are shown in Figure 3. Among all breeds, Holstein recorded the highest average r^2^ and Sistani, Sahiwal, and Nellore showed almost the same average r^2^. Average D and D′ values based on total SNPs for each autosomal chromosome in genome are shown in Figure 4 (their standard deviations are listed in Appendix A). For every breed, the pattern of LD varied dramatically across different chromosomes. For Hereford, the highest average r^2^ was found between the adjacent SNPs on chromosome 10 (0.37) and the lowest was on chromosome 28 (0.26). Considering Holstein, the highest average was on chromosome 7 (0.30) and the lowest on chromosome 27 (0.20). For Sistani, the highest was on chromosome 19 (0.32) and the lowest recorded on chromosome 6 (0.19). For Nellore, the highest was on chromosome 8 (0.29) and the lowest in chromosome 27 (0.20). Sahiwal showed the highest average on chromosome 9 (0.29) and the lowest on chromosome 28 (0.18). The Hereford breed had the highest average r^2^ (0.31) between adjacent SNPs and Sahiwal had the lowest average r^2^ (0.24) between adjacent SNPs.

All cattle breeds’ average r^2^ values and standard deviations at various physical distances, including 1–50 Kb, 50–100 Kb, 100–500 Kb, and 0.5–1 Mb, in autosomal chromosomes were calculated. The statistical data for average r^2^ are displayed in Table 2 as the distance between SNP pairs up to 1 Mb in each breed. Herford had the greatest r^2^ value (0.435) while Sahiwal had the lowest value (0.279) for segments less than 100 Kb, or lengths between 1 and 50 Kb and 50 and 100 Kb. Sistani had the greatest average r^2^ value (0.211) and Holstein had the lowest value (0.157) at a distance of 100–500 Kb. Holstein had the greatest average r^2^ value (0.035) while Sahiwal had the lowest value (0.018) at a distance of 0.5–1 Mb.

For each breed, the average r^2^ values ++varied were different on different autosomal chromosomes. In the Sistani, Nellore, Sahiwal, Holstein, and Herford breeds, greater values of LD were discovered for BTA7 (r^2^ = 0.23), BTA22 (r^2^ = 0.26), BTA27 (r^2^ = 0.21), BTA14 (r^2^ = 0.15), and BTA12 (r^2^ = 0.13), respectively (Figure 5). This might be explained by the fact that various selection conditions in each breed have shaped the specific QTLs on various chromosomes.

All breeds faced a decrease in average r^2^ when the physical distance between markers increased. Nonetheless, different populations experienced variations to varying degrees. The average of LD at 1 Kb recorded 0.36. The highest average was for Holstein with 0.39 and the lowest was recorded in Sistani (0.33). The average r^2^ of 10 Kb recorded 0.19. The highest average at this length was for Holstein (0.22) and the lowest was for Herford (0.17). The average r^2^ of 100 Kb was 0.11. At this distance, the highest average was for the Sistani breed (0.14) and the lowest was for Hereford (0.06). The Hereford breed showed faster LD reduction than the other breeds at all distances.

## 4. Discussion

Typically, r^2^ statistics are a better scale for measure of LD than D′ [29]. Trends in r^2^ can indicate that how many markers or progeny are required for QTL identification in a genome study [29]. In particular, the sample size for QTL detection must increase to identify markers which linked to QTL [3,30]. Although D′ may not be highly effective in predicting the marker density required for genomic research based on LD, its consideration of allele frequencies provides a significant advantage in detecting LD signals, especially for rare variants where allele frequencies are inherently low. The second reason for using r^2^ for measuring LD compared with D′ is that when the sample size is small or allele frequency is low, amounts of D′ become very high [3,30]. In research carried out using three statistics, namely, r^2^, D′, and D, it has been shown that by increasing the physical distance between SNPs, the average obtained for each of these statistics was reduced. However, the changes between different chromosomes were not similar [3]. This may be related to the rate of recombination and various selection pressures on loci on different chromosomes.

In our research, the observed average r^2^ values were significantly different between indigenous cattle form the Sistani province and Bos taurus and Bos indicus breeds. Different population histories, selection pressures, and amount of inbreeding within each breed may be the cause of this. LD was estimated with the three methods of D, D′, and r^2^. The estimated value of LD decayed with the increase in distance between SNP pairs. The reduction in r^2^ was faster than in D and D′ [12]. In our study, the decline rate of r^2^ was also faster compared to the statistics D and D′.

The LD trend was also examined in our study for distances ranging from 1 Kb to 1 Mb. Hereford had the greatest r^2^ value (0.435) while Sahiwal had the lowest value (0.279) for segments less than 100 Kb, or lengths between 1 and 50 Kb and 50 and 100 Kb. Sistani had the greatest average r^2^ value (0.211) and Holstein had the lowest value (0.157) at a distance of 100–500 Kb. Holstein had the greatest average r^2^ value (0.035) while Sahiwal had the lowest value (0.018) at a distance of 0.5–1 Mb. When the patterns of LD decay from various breeds were compared, it was found that, at varying distances, the Sistani animals’ pattern resembled that of the Nellore and Sahiwal cattle. Salomon-Torres et al. [31] explored 19 different cattle breeds’ LD values. According to their findings, Piedmontese (0.085), Sheko (0.104), and Charolais (0.105) had the lowest LD values for lengths ranging from 95 to 100 KB, while Hereford (0.222) and Jersey (0.201) had the highest LD averages. According to Khatkar et al. [12], research using very small sample sizes is vulnerable to bias and accuracy loss, and the degree of this bias may vary with the distance between markers. Undoubtedly, the low sample sizes affected the r^2^ values that we discovered for the different breeds of cows in our study.

In our research, we observed different trends in LD decay on different chromosomes. Different selection pressures on QTLs dispersed across the genome might result in different LD patterns on specific chromosomes across different breeds. Moreover, significant LD variability was seen between the various chromosomes, which is likely a sign of variable recombination rates, selection pressures, and genetic drift between chromosomes [32]. Therefore, chromosomes containing QTLs under selection are predicted to have greater LD [2,30]. The density of SNP panels, sample size, MAF thresholds, and distance markers have all been shown to have an impact on LD levels in several studies [6,33]. It would be realistic to assume considerable LD in Iranian cattle populations notwithstanding the sample size issues. This study’s primary concept was centered on comparing the LD values using a broad image between Sistani native cattle and comparing it with taurine and indicine breeds. In various breeds, greater changes in LD levels were observed (at <10 Kb distance). In previous studies, the LD levels in indicine breeds were smaller than in taurine breeds [33]. In the present study, Sahiwal and Nellore (indicine), compared with Hereford and Holstein breeds (taurine), had a lower LD rate. According to Meuwissen et al. [34], in order to obtain an accuracy of 0.85 for genomic breeding values, the LD levels should be higher than 0.2. It has been proposed that useful LD should be greater than 0.3 to provide enough statistical power for GWAS [35,36]. In our research, average r^2^ more than 0.2 was recorded at distances less than 100 Kb for all breeds and between 100 and 500 Kb, r^2^ above 0.2 was recorded for Sistani, Nellore, and Sahiwal. Moreover, average r^2^ higher than 0.3 was recorded at distances less than 50 Kb for all breeds, while this amount for Holstein and Herford was observed at distances between 50 and 100 Kb. Consequently, our results indicate that the densities of SNPs utilized will produce sufficient accuracy for the genomic selection in the population and for GWAS provided that a sufficient number of individuals are phenotyped for the traits of interest.

A considerable number of the SNPs in this study were removed from the dataset because they were of insufficient quality. Although the GC-score, which indicates the quality of genotyping, did not show any significant deviations (GC scores were less than 0.5 for 14% of markers), we do not believe there have been any issues with the quality of the DNA. Moreover, after measuring the quality scores of sequences for 10 randomly chosen cattle, we noted the dependability of the genotypes in order to carry out genomic analysis in our work. In the present study, the influence of SNP ascertainment bias must be acknowledged. Since the reference genome is also from a taurine breed, genotyping quality may have been somewhat impacted. Additionally, European taurine cow data were mostly used to identify the SNPs on the chip, leaving Iranian indigenous animals out of the bovine genome sequencing programs. Still, compared to 50 k chips, the remaining SNPs were sufficient for investigation and might yield more information. These findings demonstrated that the SNP density employed in this study can offer sufficient precision for future GWAS and genetic selection initiatives.

## 5. Conclusions

In the present study, LD values accurately represented both a genetic similarity and varying historical backgrounds in different cattle breeds. The Sistani breed has an LD decay like indicine cattle (Nellore and Sahiwal) and this similarity likely is due to the close distance between the Sistan and Baluchestan province and Pakistan, which is the origin of indicine breeds, or they might have the same ascertainment bias on the chip used in the assessment. The SNP density used in this study will enable the implementation of genomic selection programs and whole-genome studies in native cattle of Iran.

## Figures and Tables

**Figure 1 animals-14-03317-f001:**
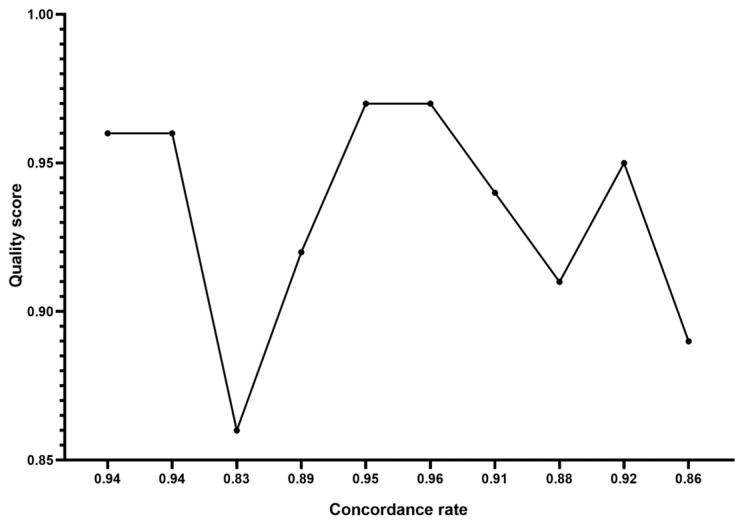
Quality score and concordance rate percentage for 10 randomly chosen samples.

**Figure 2 animals-14-03317-f002:**
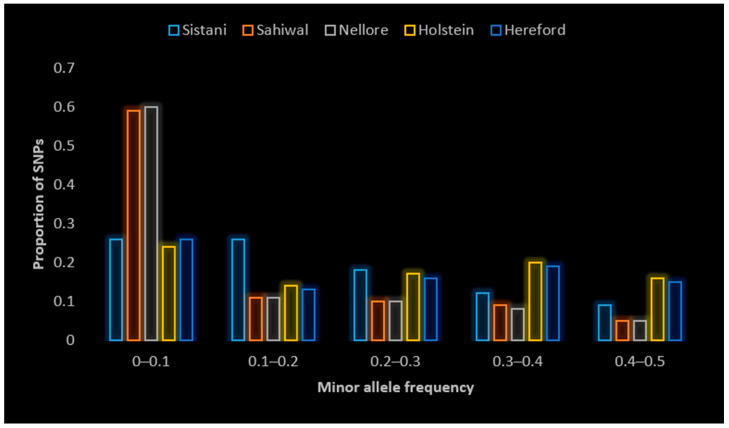
Distribution of minor allele frequencies in different breeds.

**Figure 3 animals-14-03317-f003:**
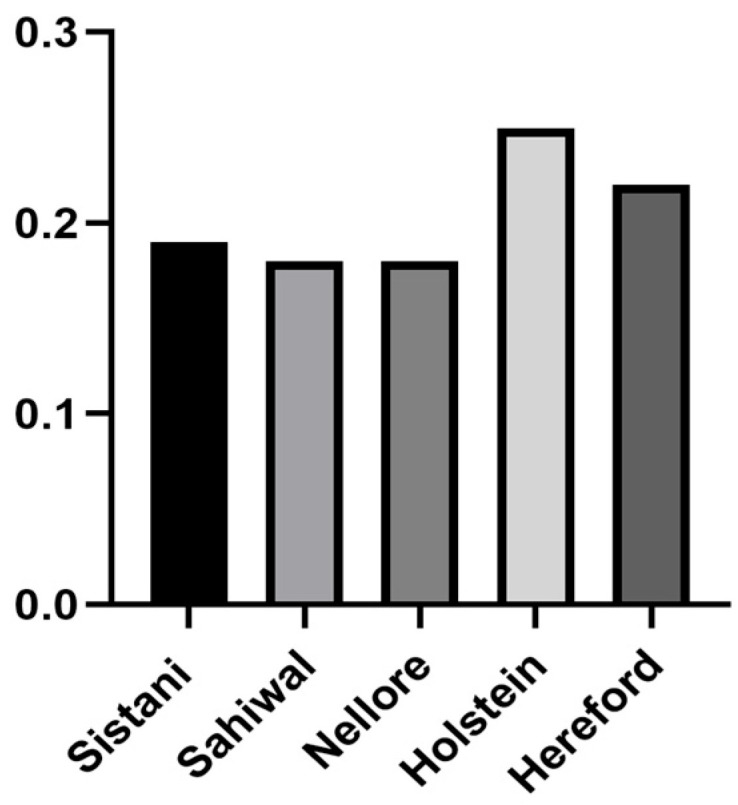
Comparison of corrected average r^2^ of all SNPs among different breeds of cattle.

**Figure 4 animals-14-03317-f004:**
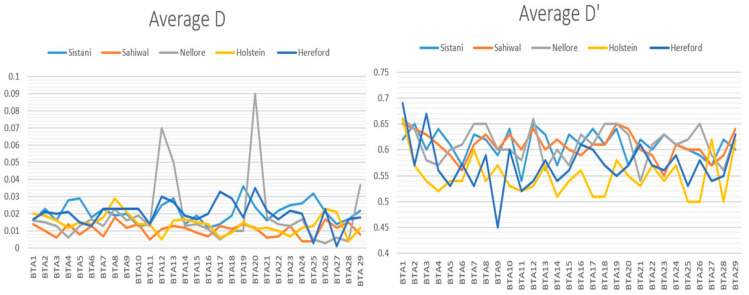
Average D and D′ based on total SNPs for each autosomal chromosome in genome in different breeds.

**Figure 5 animals-14-03317-f005:**
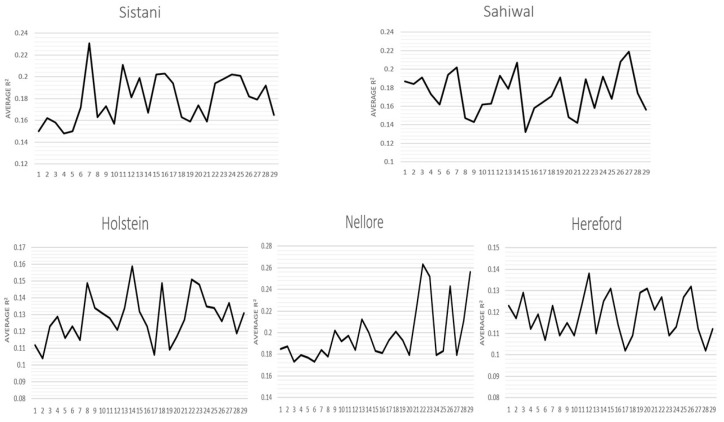
Comparison of the average r^2^ values for every chromosome across all cattle breeds.

**Table 1 animals-14-03317-t001:** Representation of total number of SNPs studied, average MAF, average SNP interval (Kb), and total number of pairwise comparisons of SNPs for each breed.

Breed	Sample Number	MAF Mean	SNP Distance (Kb)	Total Analyzed SNPs	Total SNP Pairwise Comparisons
Sistani	12	0.213	218	132,111	4,025,322
Hereford	12	0.241	137	153,101	4,704,312
Holstein	12	0.252	127	168,143	5,303,145
Nellore	12	0.130	197	143,211	4,502,119
Sahiwal	12	0.132	198	167,657	5,021,453

**Table 2 animals-14-03317-t002:** All cattle breeds’ mean r^2^ values at various physical distances, summed across all autosomes.

Distance Between SNPs	Breed	Average r^2^ ± Standard Deviation
1–50 Kb	Sistani Nellore Sahiwal Holstein Hereford	0.367 ± 0.372 0.372 ± 0.335 0.369 ± 0.341 0.419 ± 0.347 0.435 ± 0.323
50–100 Kb	Sistani Nellore Sahiwal Holstein Hereford	0.285 ± 0.326 0.298 ± 0.341 0.279 ± 0.302 0.315 ± 0.345 0.302 ± 0.331
100–500 Kb	Sistani Nellore Sahiwal Holstein Hereford	0.211 ± 0.234 0.202 ± 0.232 0.205 ± 0.212 0.157 ± 0.182 0.165 ± 0.221
0.5–1 Mb	Sistani Nellore Sahiwal Holstein Hereford	0.022 ± 0.059 0.020 ± 0.051 0.018 ± 0.048 0.035 ± 0.065 0.032 ± 0.061

## Data Availability

Some datasets presented in this article are not readily available due to technical limitations. However, some datasets are included in the Appendix A and more other datasets are available upon request.

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
