# Peer review of "Linkage Disequilibrium Decay in Selected Cattle Breeds"

_animals, 2024, doi:10.3390/ani14223317_

Round 1

Reviewer 1 Report

Comments and Suggestions for Authors

Linkage disequilibrium decay in selected cattle breeds

The study conducted a comprehensive analysis of linkage disequilibrium (LD) across various cattle breeds, utilizing genotyping data from 60 individuals of Bos indicus, Bos Taurus, and the indigenous Sistani cattle, obtained through the Illumina Bovine HD 770k chip. After applying stringent quality control measures and excluding SNPs with low minor allele frequency and Hardy-Weinberg equilibrium deviations, the researchers measured LD using D' and r2 statistics at different genomic distances. They discovered that the Sistani breed exhibited an LD decay pattern akin to Indicine cattle, potentially due to geographical proximity to the Indicine breed's origin or shared biases in the genotyping chips. The study concluded that the SNP density was adequate for genomic selection and whole-genome association studies in Iranian native cattle, providing valuable insights into the genetic structure and historical backgrounds of the breeds analyzed.Adequate workload supports the article content well, my opinion is modify. But there are some issues need to be improved, and my comments are as follows:

1.In the abstract,there isthis similarity might happen due to close distance between Sistani-Baluchistan province to Pa- kistan which is the origin of Indicine breed.Is it possible to rephrase it as: "Sistani-Baluchistan cattle, due to their proximity to cattle in Pakistan, may have had genetic or kinship relationships over a long historical period"?

2.In the introduce, it is mentioned that the indigenous cattle in Sistan-Baluchistan province have a long history, with complex bloodlines and genetic relationships. Is it necessary to experimentally prove a minimal genetic relationship with the other two subspecies of cattle?

3.In Material and Methods,Should the collection locations of Sahiwal (dairy), Nellore (beef), Hereford (beef), and Holstein (dairy) cattle be specified, as well as their relationship with the indigenous cattle in Sistan-Baluchistan province?

4.The concordance rate is calculated by comparing the degree of match between genotype data and actual data. The QC steps include excluding low-quality SNP sites, low-frequency alleles, and sites that deviate from Hardy-Weinberg equilibrium. The formulas provided in the paper require further explanation.

5.Indicate the meaning of each color and node in Figure 1.

Materials and methods

6.Can the background of Figures 3 and 5 be changed to a light color, and can the bars be changed to solid dark colors distinguished by different shades?

7.Could Figure 5 be changed to a line graph to more clearly reflect the data changes?

8.Could the curves in Figure 4 be made thinner to more clearly indicate the inflection points?

9.In Table 2, specify the Distance Deviations situation for each BREED.

10.While the original statement mentions that "the D' has no great ability to predict the marker density needed for genomic research based on LD," it's important to note that D' takes allele frequencies into account, which allows it to capture LD signals even when allele frequencies are low. This is particularly important in the study of rare variants. A revised way to express this could be:"Although D' may not be highly effective in predicting the marker density required for genomic research based on LD, its consideration of allele frequencies provides a significant advantage in detecting LD signals, especially for rare variants where allele frequencies are inherently low."

11.It is recommended to supplement the data on the GC scores of the samples or other evidence of DNA quality.

12. Is it possible to conduct an analysis to verify the LD variation among different breeds or design methods to mitigate the impact of not including Iranian native animals in the chip samples?

.

Author Response

Dear Respected Reviewer 1

We would like to extend our deepest gratitude to you due to your great comments and suggestions. Undoubtedly, your suggestions and requests put this paper in higher level and more worthy for publication. Please see our answers to your suggestions and requests. We tried our best to fulfill your requirements and hope we have done this job as you wish.

Comment 1: 1.In the abstract there is“this similarity might happen due to close distance between Sistani-Baluchistan province to Pa- kistan which is the origin of Indicine breed”.Is it possible to rephrase it as: "Sistani-Baluchistan cattle, due to their proximity to cattle in Pakistan, may have had genetic or kinship relationships over a long historical period"?

Response: Thank you so much for your comments. It was an overlap review from your kind side and different reviewer. We try to include both of your suggestions together to address both reviewers comment. Please see line number 30 to 34.

Comment 2: In the introduce, it is mentioned that the indigenous cattle in Sistan-Baluchistan province have a long history, with complex bloodlines and genetic relationships. Is it necessary to experimentally prove a minimal genetic relationship with the other two subspecies of cattle?

Response: Thank you so much for your comment. Actually we did not understand which part of introduction that we have meant complex bloodlines and genetic relationships of Sistani breed. As the matter of fact, Sistani breed is an indigenous breed of Sistan and Baluchestan area which has been adapted to hard climate of this province. In second, the scope of our Analysis is mostly quantitative analysis of LD with almost limited resources. But we undoubtedly use your kind idea in our next experimental project.

Comment 3: In Material and Methods,Should the collection locations of Sahiwal (dairy), Nellore (beef), Hereford (beef), and Holstein (dairy) cattle be specified, as well as their relationship with the indigenous cattle in Sistan-Baluchistan province?

Response: Thank you so much for your comment. Modified. Please see line 93 and 94. To clarify that, Sistani, Sahiwal and Nellore are basically belong to the common region that might lead them to have the same pool of kinship. In the past Sistan and Panjub belong to the same region, however it seems that there is no relation of kinship between these breeds with Hereford and Holstein.

Comment 4: The concordance rate is calculated by comparing the degree of match between genotype data and actual data. The QC steps include excluding low-quality SNP sites, low-frequency alleles, and sites that deviate from Hardy-Weinberg equilibrium. The formulas provided in the paper require further explanation.

Response: Thank you so much for your comment. Modified. Please see line 102 to 105 and before Estimation of linkage disequilibrium subtitle.

Comment 5: Indicate the meaning of each color and node in Figure 1.

Reponses: Thank you so much for your comment. The color of plot and background belongs to the design of graph created in Graphpad prism software. Therefore, they (the color) don’t convey any meaning otherwise it would be explained completely. However, to fulfill your kind requirement, we change the white back ground with black bars for stopping any misunderstanding.

Please see figure 1. Line 149.

Comment 6: Can the background of Figures 3 and 5 be changed to a light color, and can the bars be changed to solid dark colors distinguished by different shades?

Response: Thank you so much for your comments. To completely fulfill your kind suggestion, we tried to improve the quality of all figures, Figures 3 and Figure 5; we change the color and bar changed to solid dark with different shades. Please see figure 3 and 5. Line 183 and and 206.

Comment 7: Could Figure 5 be changed to a line graph to more clearly reflect the data changes?

Response: Thank you so much for your comment. Modified. Please see figure 5. Line 206.

Comment 8: Could the curves in Figure 4 be made thinner to more clearly indicate the inflection points?

Response: Thank you very much for your comment. Modified. Please see Figure 1. Line 185.

Comment 9: In Table 2, specify the Distance Deviations situation for each BREED.

Response: Thank you so much for your comment. Firstly for distance deviation we basically meant distanced between SNPs so we changed the headline. For distance deviation situation, the distance classes are the same for all breeds. That’s why did not mention that in the table.

Comment 10: While the original statement mentions that "the D' has no great ability to predict the marker density needed for genomic research based on LD," it's important to note that D' takes allele frequencies into account, which allows it to capture LD signals even when allele frequencies are low. This is particularly important in the study of rare variants. A revised way to express this could be:"Although D' may not be highly effective in predicting the marker density required for genomic research based on LD, its consideration of allele frequencies provides a significant advantage in detecting LD signals, especially for rare variants where allele frequencies are inherently low."

Response: Thank you so much. Revised. Please see line 219-222.

Comment 11: It is recommended to supplement the data on the GC scores of the samples or other evidence of DNA quality.

Response: Thank you so much for your comment. Added. Please see the supplementary material in line 306.

Comment 12: Is it possible to conduct an analysis to verify the LD variation among different breeds or design methods to mitigate the impact of not including Iranian native animals in the chip samples?

Response: Thank you so much for your Comment. LD variation among different breeds would be analyzed via large and deep pedigrees for the populations investigated. Regrettably, at the moment, we don’t have such information. But as already mentioned, the sooner we get complete pedigree information we will use your amazing idea in our next project.

Reviewer 2 Report

Comments and Suggestions for Authors

This study examines the linkage disequilibrium (LD) patterns in 60 cattle from 5 breeds using Bovine HD SNP array data. The authors conclude that the LD pattern of the indigenous Sistani cattle is more similar to Bos indicus than Bos taurus. While the topic and design are sound, and the scientific writing is adequate but requires more careful attention to formatting, here are my suggestions for improvement:

In line 20, the phrase "Bos Taurus and indicus" should be corrected to "_Bos taurus_ and _Bos indicus_" for consistency with species nomenclature. This should be revised throughout the paper.

In line 30, the sentence "In the present study, Sistani breed have a LD decay like Indicine cattle (Nellore and Sahiwal)..." is awkward. A more precise version could be: "In this study, the Sistani breed shows LD decay patterns similar to Indicine cattle (Nellore and Sahiwal), which may be due to the geographic proximity of the Sistani-Baluchistan province to Pakistan, the origin of Indicine breeds, or due to ascertainment bias in the SNP chips used." It would also be beneficial to include an LD decay plot to show the decline in LD as the distance between markers increases.

In line 93, the phrase "as well as indigenous cattle in Sistan-Baluchistan province" should be clarified to "as well as indigenous cattle (Sistani) from the Sistan-Baluchistan province" to make it more specific.

In line 107, the method used for calculating D' should be cited, and the term "A1_B1" needs to be explained in more detail for clarity.

In line 123, it is mentioned that the statistical analysis was performed using GraphPad Prism and Excel, with a significance criterion of P < 0.05. However, it would be helpful to discuss the choice of SnppldHD for LD calculation, particularly in comparison to more commonly used tools like PLINK or VCFtools. Explaining the differences or advantages of SnppldHD over these tools would enhance the clarity of the methodology.

In line 125, "P < 0.05" and in line 130, "p-value <10−7" use inconsistent formatting for the term "P-value." The format should be standardized throughout the paper.

In line 133, the phrase "in each breed The general QC" is missing a period after "breed" and should be corrected to "in each breed. The general QC."

In line 140, "Average of MAF" should be revised to "average of MAF" for consistency in capitalization.

In line 153, "Minor" should be changed to "minor" to follow proper capitalization rules.

Author Response

Dear Respected Reviewer 2

We would like to extend our deepest gratitude to you due to your great comments and suggestions. Undoubtedly, your suggestions and requests put this paper in higher level and more worthy for publication. Please see our answers to your suggestions and requests. We tried our best to fulfill your requirements and hope we have done this job as you wish.

Comment 1: In line 20, the phrase "Bos Taurus and indicus" should be corrected to "_Bos taurus_ and _Bos indicus_" for consistency with species nomenclature. This should be revised throughout the paper.

Response: Thank you so much for your comment. Modified. Please see line 20 and 230.

Comment 2: In line 30, the sentence "In the present study, Sistani breed have a LD decay like Indicine cattle (Nellore and Sahiwal)..." is awkward. A more precise version could be: "In this study, the Sistani breed shows LD decay patterns similar to Indicine cattle (Nellore and Sahiwal), which may be due to the geographic proximity of the Sistani-Baluchistan province to Pakistan, the origin of Indicine breeds, or due to ascertainment bias in the SNP chips used." It would also be beneficial to include an LD decay plot to show the decline in LD as the distance between markers increases.

Response: Thank you so much for your comments. It was an overlap review from your kind side and different reviewer. We try to include both of your suggestions together to address both reviewers comment. Please see line number 30 to 34.

Comment 3: In line 93, the phrase "as well as indigenous cattle in Sistan-Baluchistan province" should be clarified to "as well as indigenous cattle (Sistani) from the Sistan-Baluchistan province" to make it more specific.

Response: Thank you so much for your response. Revised. Please see line 92 and 93.

Comment 4: In line 107, the method used for calculating D' should be cited, and the term "A1_B1" needs to be explained in more detail for clarity.

Response: Thank you very much for your comment. Refrence added as well as more explanation. Please see line 107 to 111.

Comment 5: In line 123, it is mentioned that the statistical analysis was performed using GraphPad Prism and Excel, with a significance criterion of P < 0.05. However, it would be helpful to discuss the choice of SnppldHD for LD calculation, particularly in comparison to more commonly used tools like PLINK or VCFtools. Explaining the differences or advantages of SnppldHD over these tools would enhance the clarity of the methodology.

Response:  Thank you very much for your comment. Please see line 128 to 130.

Comment 6: In line 125, "P < 0.05" and in line 130, "p-value <10−7" use inconsistent formatting for the term "P-value." The format should be standardized throughout the paper.

Response: Thank you so much for your comment. Revised through the manuscript. Please see line 99 and line 135.

Comment 7: In line 133, the phrase "in each breed The general QC" is missing a period after "breed" and should be corrected to "in each breed. The general QC."

Reponse: Thank you so much for your comment. Actually in these breed and the general QC belong to two different sentences separated by dot. Please see line 139.

In line 140, "Average of MAF" should be revised to "average of MAF" for consistency in capitalization.

Response: Thank you so much for your comment. We removed average because of similarity of this sentence with other researches based on the editor comment.

Comment 8: In line 153, "Minor" should be changed to "minor" to follow proper capitalization rules.

Response: Thank you very much. Revised. Please see line 158.

Reviewer 3 Report

Comments and Suggestions for Authors

The revision of the manuscript “Linkage disequilibrium decay in selected cattle breeds”

Mayor comments.

In general, the manuscript is well written. The figure quality must be improved, with a clearer combination of color, etc.

The main concern about the manuscripts is that the topic is well-studied in different breeds. Novel knowledge is not clear at all.

The authors must review the real contribution to the knowledge, my advice is chance to research note.

Author Response

Dear Respected Reviewer 3

We would like to extend our deepest gratitude to you due to your great comments and suggestions.

Comment 1: In general, the manuscript is well written. The figure quality must be improved, with a clearer combination of color, etc.

Response: Thank you so much for your comment. Based on your comment. We increased the quality of all figures and make them more readable.

Comment 2; the main concern about the manuscripts is that the topic is well-studied in different breeds. Novel knowledge is not clear at all.

Response: Thank you very much for your concern. Well the purpose of our analysis was to check whether The Sistani breed had any similarity with compared breeds. We tried as much as possible to do data analysis as precise as possible. The single nucleotide polymorphism (SNP) density used in this study will enable implementation of genomic selection programs and whole genome studies in native cattle different region of Middle East.

Comment 3: The authors must review the real contribution to the knowledge, my advice is chance to research note.

Response: Thank you so much for your comment. We have already tried that and included good research related notes in our manuscript including those literature related to calculating LD. However if any specific place in manuscript need more revision please let us know.

Round 2

Reviewer 3 Report

Comments and Suggestions for Authors

I agree with the revised version.